# Cochineal (*Dactylopius coccus* Costa) Pigment Extraction Assisted by Ultrasound and Microwave Techniques

**DOI:** 10.3390/molecules29235568

**Published:** 2024-11-25

**Authors:** Rogelio Reyes-Pérez, Juanita Pérez-Hernández, Minerva Rosas-Morales, Miguel Ángel Plascencia-Espinosa, Oxana Lazo-Zamalloa, Valentín López-Gayou, Pedro Antonio López, Gabriel Ríos-Cortés, Ada María Ríos-Cortés

**Affiliations:** 1Centro de Desarrollo de Productos Bióticos, Instituto Politécnico Nacional, Carr Yautepec-Jojutla s/n-Km. 85, San Isidro 62739, Morelos, Mexico; rreyesp@ipn.mx; 2Centro de Investigación Biomédica del Sur, Instituto Mexicano del Seguro Social, C. Rep. Argentina 1, Xochitepec 62790, Morelos, Mexico; kika_juanita@yahoo.com.mx; 3Centro de Investigación en Biotecnología Aplicada, Instituto Politécnico Nacional, Ex-Hacienda San Juan Molino Carretera Estatal Tecuexcomac-Tepetitla Km. 1.5, Tepetitla de Lardizábal 90700, Tlaxcala, Mexico; mrosasmo@ipn.mx (M.R.-M.); mplascencia@ipn.mx (M.Á.P.-E.); olazoz@ipn.mx (O.L.-Z.); vlopezg@ipn.mx (V.L.-G.); 4Colegio de Postgraduados, Campus Puebla, Boulevard Forjadores de Puebla No. 205, Santiago Momoxpan, Municipio de San Pedro Cholula 72760, Puebla, Mexico; palopez@colpos.mx; 5Departamento de Ingeniería Química, Tecnológico Nacional de México, Instituto Tecnológico de Orizaba, Oriente 9, Orizaba 94320, Veracruz, Mexico

**Keywords:** natural pigment, ultrasound, microwave, green extraction, Box–Behnken design

## Abstract

Carminic acid is a natural pigment typically found in several insect taxa, including specific insects such as “grana cochinilla fina” in Mexico (*Dactylopius coccus* Costa). Commercially, it is also referred to as carmine, which is a more concentrated solution presenting as at least 50% carminic acid. To date, this dye has been used in the pharmaceutical, food and cosmetic industries. Unfortunately, one of the main limitations has to do with establishing the appropriate extraction and purification protocol. Currently, there is growing interest in developing eco-friendly and efficient pigment extraction processes for various applications. In this study, we compare the ultrasound- and microwave-assisted extraction versus with a conventional method to obtain carminic acid from cochineal. To do this, we considered three factors that influence the extraction process as independent variables: solvent volume, temperature and irradiation time. The optimization was carried out using the response surface methodology, employing a three-factor and three-level Box–Behnken experimental design. Carminic acid contents were quantified by UV–Vis spectroscopy, and extracts were evaluated by infrared spectroscopy to verify the integrity of the carminic acid molecule. The yield obtained by ultrasound-assisted extraction was 49.2 ± 3.25, with an efficiency of 31.3 mg/min, while microwave-assisted extraction showed a yield of 40.89 ± 2.96, with an efficiency of 27.3 mg/min. Both methods exceeded the extract yield (31.9 ± 3.4%) and efficiency (10.6 mg/min) obtained with the conventional method, demostrating that ultrasound- and microwave-assisted extraction are viable alternatives for obtaining carminic acid, with the potential to be scaled up to an industrial level.

## 1. Introduction

Carminic acid (CA) is a natural dye obtained from the female insects of *Dactylopius coccus* Costa, a parasite of the prickly pear cactus (*Opuntia ficus*-indica), also known as “grana”, “grana cochinilla fina” and “cochinilla” in México [1]. The molecular structure of CA consists of a polyhydroxylated anthraquinone with a C-glycosil side chain with many hydroxyl moieties, as shown Figure 1. Therefore, this dye shows solubility in different media, such as water, alcohol, esters and acidic and alkaline solutions. This dye is a deep red pigment that produces other color schemes from orange to violet under varying pH [2,3].

CA is one of the oldest natural dyes widely used as red colorants approved by the FDA, and it is included in the list of additives of the European community (E120) for use in pharmaceuticals in foods, beverages, cosmetics and textiles due to its high chemical and biological stability [4,5]. Moreover, several studies have reported that CA acts as an antioxidant against free radicals and reactive oxygen species (ROS), such as singlet oxygen [6,7,8].

According to González et al. (2002) [9], cochineal also contains other less commonly used pigments that are only analyzed to determine if there are adulterations when forming the carmine lake, and that which corresponds to the lakes is obtained by the union of combining carminic acid with salts such as aluminum or calcium [10].

The extraction of natural pigments is complicated because many factors are involved that could alter the pigmentation properties and color stability. The selection of suitable extraction technology is of utmost importance, as the selected method must be efficient and economical while ensuring pigment yield and stability. Conventional extraction methods include simple (solid–liquid), inexpensive and easy-to-use techniques; these processes may include treatment with organic solvents, alkaline extraction, solid–liquid separation, the formation of insoluble lakes, recovery by centrifugation, the resolubilization of carminic acid and concentration [10]. These traditional extraction techniques have certain limitations, such as high energy consumption, high solvent usage, long extraction times and low pigment yield, which hamper their efficiency and sustainability, resulting in higher production costs [9,10,11].

Different conventional extraction methods have been developed and used to obtain this pigment at the industrial level [12]. To solve these problems at the different extraction stages, green extraction methods are emerging technologies, including supercritical fluid extraction, microwave-assisted extraction (MAE), ultrasound-assisted extraction (UAE) and high-pressure homogenization, which involve new conceptualization and the design of extraction processes and solvent applications to eliminate the use and generation of dangerous substances. These technologies demonstrated the capability of being a sustainable alternative to conventional extraction, showing the potential to increase the extraction yield and decrease the extraction time, energy and solvent consumption [13,14,15].

Ultrasound-assisted extraction (UAE) is based on cavitation defined as the formation, growth and collapse of microbubbles in the extraction solvent pushing to the processing material, which promotes a higher contact and enhances the mass transfer by the solvent moving inward and outward from the material [16,17]. Microwave-assisted extraction (MAE) is based on the distribution of microwave energy on the material to be processed, penetrating inside and interacting with the polar components to generate heat, acting directly on the molecules by ionic conduction and dipolar rotation, which allows for the selective heating of samples as a function of its dielectric constant [18,19]. These two techniques have stood out for their efficiency in extracting natural pigments [20,21].

Therefore, the aim of this work was establishing the optimal extraction conditions by ultrasound and microwave techniques to enhance production yields and efficiency, lowering extraction times specifically of carminic acid.

To evaluate such conditions, a surface response methodology was applied based on a mathematical model for extraction optimizations. It used a Box–Behnken design to optimize diverse parameters such as the solvent amount, processing time and temperature in order to evaluate different variables and interaction effects [22,23]. The quantification of cochineal pigments was made by UV spectrophotometry, and the assessment of the carminic acid molecule was performed by FTIR (Fourier Transform Infrared), verifying that it remained unaffected after these processes.

## 2. Results

### 2.1. Cochineal Extraction Yield Obtained by Conventional Method

Cochineal pigment was extracted by the conventional method (CM), obtaining an extract yield of 31.9 ± 3.4% within 30 min, with an efficiency of 10.6 mg/min.

### 2.2. Fractional Factorial Design

The results of the fractional factorial design test are displayed in Table 1

### 2.3. Statistical Analysis

The experimental design allowed us to identify the significative factors that had an influence on the yield of the extractions. An ANOVA test was performed for each response (Table 2), and the obtained results indicate that carminic acid yields obtained by ultrasound showed a significant difference in the solvent and temperature interactions (*p* < 0.05). For the case of the ANOVA analysis of microwave extractions, the results indicate there is a significant difference only at the solvent factor (*p* < 0.05).

### 2.4. Analysis of Response Surface Plots

#### 2.4.1. Extraction Assisted by Ultrasound

The results obtained show the response surface and surface contour plots for the yields achieved by UAE (Figure 2). In Figure 2b, a maximum extraction yield of 49.2 ± 3.25% is observed at a temperature of 60 °C for 15 min, with a solvent ratio of 1:20 g/mL. The contour plot suggests that this yield can be reached in a range of 13 to 15 min and between 60 °C and 69 °C, maintaining the same solvent ratio (1:20 g/mL). These results highlight a 6.8% improvement in extraction yield compared to that obtained by Pressurized Liquid Extraction (PLE; 42.4%) and 9.8% higher than that reported with supercritical fluid extraction (SFE; 39.4%), as reported in previous studies by Borges et al. [24].

The average levels (Box–Behnken) of 70 °C, 10 min and 15 mL of solvent were considered, from which the estimated response has a change with a minor modification in the experimental factors. The optimal variables were obtained for the UAE method, predicting that the optimal temperature was 63.2 °C at 15.0 min with a sample/solvent ratio of 1:20 g/mL, reaching an extract yield of 50.3%, with an increase of 1.1% with respect to the estimated value.

#### 2.4.2. Extraction Assisted by Microwaves

The analysis of variance performed for the obtained results from MAE indicates there is only a significant difference for the solvent factor (*p* < 0.05).

The highest extract yield was 41.0 ± 1.04% for MAE at 60 °C at 15 min, with a solvent ratio of 1:20 g/mL. Surface contour plots show the variation in yield in relation to the three factors analyzed (Figure 3), showing a yield 1.4% lower than that obtained by PLE and 0.6% higher than that reported by Borges et al. by SFE [24].

The optimal variables were obtained for the MAE method, predicting that the optimal temperature is 60 °C at 15.0 min, with a sample/solvent ratio of 1:20 g/mL, reaching a yield of 41.00%, with an increase of 1.1% from the estimated value.

### 2.5. Comparison Between the Conventional Method and Green Extraction Techniques

Table 3 presents a detailed comparison of the extract yields, carminic acid contents and efficiencies obtained for the different methods evaluated. As observed, conventional extraction showed the lowest yield, reaching 17.8%, with an efficiency of 2.0 mg/min of CA. In contrast, UAE demonstrated higher yields compared to the green extraction methods (PLE 42.4% and SFE 39.4%) reported by Borges et al. [24].

Table 3 shows the comparison between the obtained yields of extract, carminic acid and efficiencies of the evaluated methods. The conventional extraction resulted in a yield of 17.8%, with an efficiency of 2.0 mg/min of carminic acid, being the one with the lowest yield and efficiency. The UAE presented the best yields compared to the green methods reported by Borges [4], obtaining 42.4% with the PLE method and 39.4% with the SFE method.

### 2.6. Infrared Spectra of Carminic Acid Extracts

A spectral analysis at the infrared region was carried out using a standard commercial control of carminic acid, which was compared to the analysis obtained by Cañamares et al. (2006) [25] and the extracts obtained by the proposed UAE and MAE methods.

Figure 4 shows the comparison of IR spectra, where the characteristic bands attributable to carminic acid are observed and compared to those reported by Cañamares et al., which are also shown in Table 4, mainly related to absorption peaks related to anthraquinones (1568 and 1273 cm^−1^), sugar residues (2933, 1074 and 1044 cm^−1^) and carboxyl groups (3302, 1616 and 1377 cm^−1^). The extraction methods proposed in this work also correlate to the bands assigned to these functional groups present from standard controls and carminic acid signals reported by Cañamares, suggesting that we are obtaining carminic acid pigments using UAE and MAE methods.

The peaks observed at 1616 and 1243 cm^−1^ are related to C=O and C=C groups from the aromatic ring of anthraquinones that are mixed and show variations, depending on each extraction method. This was corroborated by UV–vis spectroscopy, which shows the presence of carminic acid isomers and additional acids.

The UV–Vis spectra of cochineal extracts obtained from different methods, which were compared to standard spectra with characteristic absorption peaks at 275, 311, 494 and 530 nm, and the extract obtained by the conventional method showed highly similar peaks. A slight difference was perceived for UAE and MAE methods showing bands at 276, 328, 494, 527 and 562 nm, and this variation could be attributed to the obtention as well as additional isomers from carminic acid, also such as flavokermesic and kermesic acid, which have been reported by González et al. 2002 [9] (Figure 5).

## 3. Discussion

The carminic acid extraction assisted by ultrasound and microwaves as green technologies provide advantages over the conventional extraction method since they do not modify the molecular composition, obtaining higher yields and efficiencies and shorter processing times.

The use of UAE produced extraction yields of 49.2 ± 3.25 compared to the methodology reported by Borges et al. [24]; it seems to be the best and most effective for the extraction of carminic acid. It must be mentioned that UAE and conventional methods used water as the extraction media, but UAE uses cycles of low- and high-pressure waves over the sonicated liquid. Within the low-pressure wave cycle, the generated bubbles implode violently with the high-pressure waves; hence, they can break materials into tiny particles, which allows for the reduction of particles size, increasing the contact area to the solvent and the produced cavitation [26]. Cavitations are formed over the surface of cochineal specimens that implode, forming “hot spots”, with an effective temperature estimated within the range between 4500 and 5000 K. Considering these values, the pressure during collapses can be inferred through the van der Waals equation to be approximately 1700 atm [27]. This released energy results in a more efficient extraction but is not enough to cause bond breaking within the carminic acid molecule.

On the other hand, the use of MAE showed an extract yield of 41.0 ± 1.04% and carminic acid yield of 18%. Although the obtained yield was slightly higher than that reported using PLE, the extraction efficiency was clearly higher at 27.3%.

This increase in efficiency is related to the dissipation factor that determines the ability of a substance to convert the electromagnetic energy to heat. In this case, the used solvent was water, with a dissipation factor of 0.123, which is considered regular enough to absorb microwaves. The cochineal insect also has a certain amount of moisture along with proteins, carbohydrates and lipids, which absorb microwaves, promoting the extraction of carminic acid. As explained before, green methods were based on the improvement of efficiency over traditional methods by physical action over the medium. This also helps to preserve the structure of carminic acid, even at elevated temperatures. Despite it being reported that ultrasound can generate new species during nanosecond lapses, this only happens in the presence of ionic compounds [28]. Although there are many speculations about the microwave effects over molecules, these could not be structurally modified or induce a chemical reaction. The explanation is based on the microwave photon, which corresponds to the frequency used in the heating systems based on microwaves, which have an approximate energy near 0.00001 eV (2.45 GHz, 12.22 cm). According to these values, the photon of microwaves is not energetic enough to break hydrogen bonds, and it is smaller than the Brownian movement, being unable to induce chemical reactions.

From the analysis of FTIR spectra from carminic acid samples obtained by UAE and MAE, there were observed characteristic peaks related to the functional groups of the molecule, as reported by Camañares et al. [25], showing slight shifts in the wavenumber that could be related to additional compounds present in the extracts. However, the typical absorption peaks were identified from the standard spectrum, suggesting that the molecule was not modified at its chemical structure. For the case presented by Borges et al. [24], the molecule was not affected by the extraction methods PLE and SFE. Moreover, they achieved high extraction yields, and the extraction time was reduced by six.

On the other hand, the work reported by Guo et al. [29] described a Box–Behnken experimental design where the time and solvent ratio were included as factors for the two techniques: microwaves and ultrasound. Since both extraction methodologies are based on distinct physical phenomena, we detected the need to establish an experimental design where they are separate, including intervals of the variables related to each of the technologies. Guo and co-workers reported an extract yield of 95%, achieving this after 10 repeated extractions combining these methods; however, they did not report if the molecule structure of carminic acid was preserved.

The advantages of green technologies like UAE and MAE are diverse because they can highly increase the yields and efficiencies of extraction, lowering energy consumption. This can be achieved by having a significant decrease of processing times and a reduction of the solvent volume to have an eco-friendly treatment [30], which allows for a decrease in the general costs, which constitute the most desirable aspects of all extraction processes, obtaining as well high yields of the interest compounds. In addition, the cavitation effect can promote the removal of biological pollutants, an essential factor for the commercialization of this pigment [31].

The present work has elucidated the optimal variables for the extraction of carminic acid by UAE using water as the only solvent; however, it is necessary to perform scaling experiments at the industrial level for verifying the obtained results. It is important to mention that the most relevant challenges of green technologies require the involvement of industries that assume the commitment of modifying their extraction procedures, replacing harmful and pollutant materials [32]. It would greatly benefit the environment if natural pigment extraction industries could implement the green technologies proposed in this work.

For this purpose, it is suggested that the further implementation of pilot-scale experiments be used for assessing the profitability of carminic acid extraction by ultrasonication.

It will be important to apply this technology for the processing of wild-type cochineal from *D. coccus*, which is abundant in Mexico; however, it is underestimated because of its low content of carminic acid, so it is considered a plague [33], in contrast to the fine grain. Finally, one important perspective is to evaluate how the effect of ultrasonication is on the allergenic proteins (high and low density) that has the carminic acid since it has been reported before that ultrasonication treatment can decrease the allergenicity, antioxidant capability and protein digestibility of shrimp [34].

## 4. Materials and Methods

### 4.1. Cochineal Specimen Data

Cochineal insects *Dactylopius coccus* C. were provided by Asociación Rojas Tepale, located at Santa María Zacatepec, a community from Juan C. Bonilla, in Puebla, México.

### 4.2. Carminic Acid Extraction by the Conventional Method (CM)

The conventional method was taken from Borges [24], which uses water as the only extraction medium. The cochineal grain was mixed with water solvent in a ratio of 1:15 g/mL, and it was boiled without stirring for 30 min; then, the mixture was decanted, and the supernatant was filtered. The precipitate was extracted twice, and the extracts were mixed and preserved under refrigeration.

### 4.3. Ultrasound-Assisted Extraction (UAE)

A Cole Parmer^®^ ultrasonication equipment (Model CP 505, Vernon Hills, IL, USA) was employed with a 1/4 microtip, 500 Watts and a 20 Khz frequency. Distilled water was used as a solvent in different ratios with the ground cochineal grain (1:10, 1:15 and 1:20 g/mL). All samples were placed in a jacketed tube coupled to a cooling system, to regulate temperature. The testing temperatures were 60, 70 and 80 °C, whereas the ultrasound exposure times were 5, 10 and 15 min.

### 4.4. Microwave-Assisted Extraction (MAE)

The cochineal grain was previously ground and placed in vials in a ratio of 1:10, 1:15 and 1:20 g/mL with deionized water. The vials were kept in peek jackets within a circular tray of the Anthon Paar microwave equipment (Synthos 3000, 1200 W, Graz, Austria). The used temperatures were similar to the AUE extraction and also the exposure times.

### 4.5. Carminic Acid Analysis by FTIR Spectroscopy

To verify that the carminic acid molecules from the obtained solid extracts were not affected during the followed processes by the AUE, MAE and Conventional Method (CM). The cochineal extracts as well as the carminic acid standard (Sigma Aldrich, St. Louis, MO, USA) were measured using a FTIR spectrometer (Bruker Vertex 70 model, Bruker Optics Corporation, Billerica, MA, USA) in the Attenuated Total Reflectance mode (ATR module with diamond crystal), with a scanning from 4000 to 400 cm^−1^ and a resolution of 4 cm^−1^.

### 4.6. Analysis and Quantification of Carminic Acid by UV–Vis Spectroscopy

Samples obtained by conventional method and by ultrasound- and microwave-assisted extraction were analyzed by UV–Vis spectroscopy (Thermo Model Evolution 600 spectrometer (Waltham, MA, USA) with an absorption interval of 190 to 700 nm) at a wavelength of 494 nm, comparing the carminic acid standard (Sigma Aldrich 96% purity). The concentration ratio for CA quantification was 5–100 ppm. The CA regression equation was (y) = 0.0129(x) + 0.0196, r^2^ = 0.9989

All extracts obtained by the conventional method and ultrasound- and microwave-assisted extraction were brought to complete dryness using a rotary evaporator at 80 °C under reduced pressure and lyophilized.

CA yields were reported as the % of carminic acid (*w*/*w*) content present in the extract [24].

### 4.7. Experimental Design and Statistical Analysis

To optimize the extraction methods, the Box–Behnken response surface design was applied with three factors and three levels for each factor: a low level (−1), medium level (0) and high level (1). The three established independent variables were temperature (X1), solvent volume (X2) and time (X3), and the response variable was the yield of carminic acid [24].

Table 5 represents the range of independent variables and their levels: high, low and central points.

#### Corroboration of Optimal Variables

The optimal variables were obtained from an estimation of the extraction and the carminic acid yields. Therefore, experimental runs were carried out with the optimal variables obtained from UAE and MAE, using the Statgraphics Centurion XVI software version 16.2.04 for a quantitative verification of the yields of extracts and carminic acid samples.

## 5. Conclusions

In the present work, we concluded that by using the ultrasound-assisted extraction method, we were able to obtain high extract yields (49.2 ± 3.25%) with 67.34 °C temperature, 1:20 g/mL of solvent and 15 min processing time as optimized conditions, with an efficiency of 32.8 mg/min. In the case of microwave-assisted extraction, a yield of 41.0 ± 1.04% was obtained at 60 °C at 15 min, with a solvent ratio of 1:20 g/mL. % but with an efficiency of 36 mg/min for carminic acid; although the yield was lower, the efficiency in obtaining it is higher than that of the ultrasound-assisted extraction. Moreover, the carminic acid molecule remained unchanged, demonstrating that ultrasound-assisted extraction and microwave-assisted extraction are excellent green technologies for obtaining carminic acid from cochineal. Because of the growing concern about the environmental impact of industrial processes, it would be relevant to test these green technologies in the production of lakes from carminic acid.

## 6. Patents

This work was based on two patent applications: MX/a/2021/015175 and Mx/a/2021/015176

## Figures and Tables

**Figure 1 molecules-29-05568-f001:**
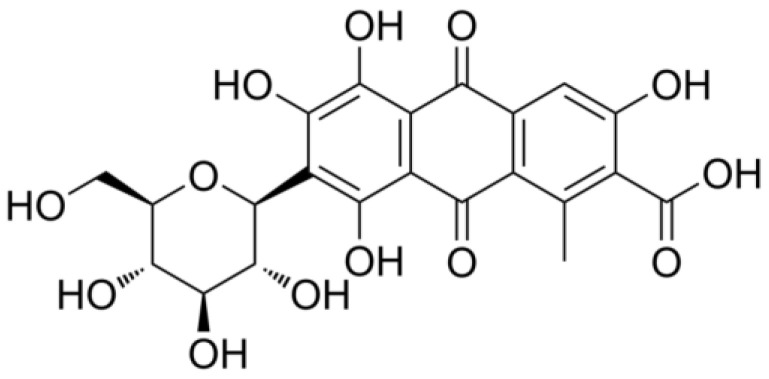
Molecular structure of carminic acid.

**Figure 2 molecules-29-05568-f002:**
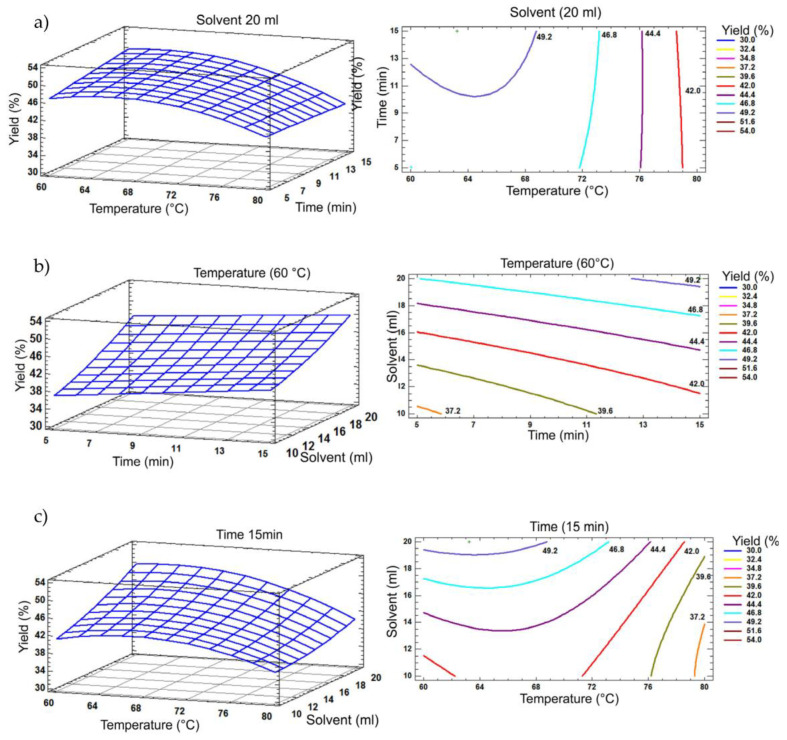
A response surface plot showing the effects of solvent, temperature and processing time obtained by UAE. The presented plots are related to (**a**) solvent (20 mL), (**b**) temperature (60 °C) and (**c**) time (15 min).

**Figure 3 molecules-29-05568-f003:**
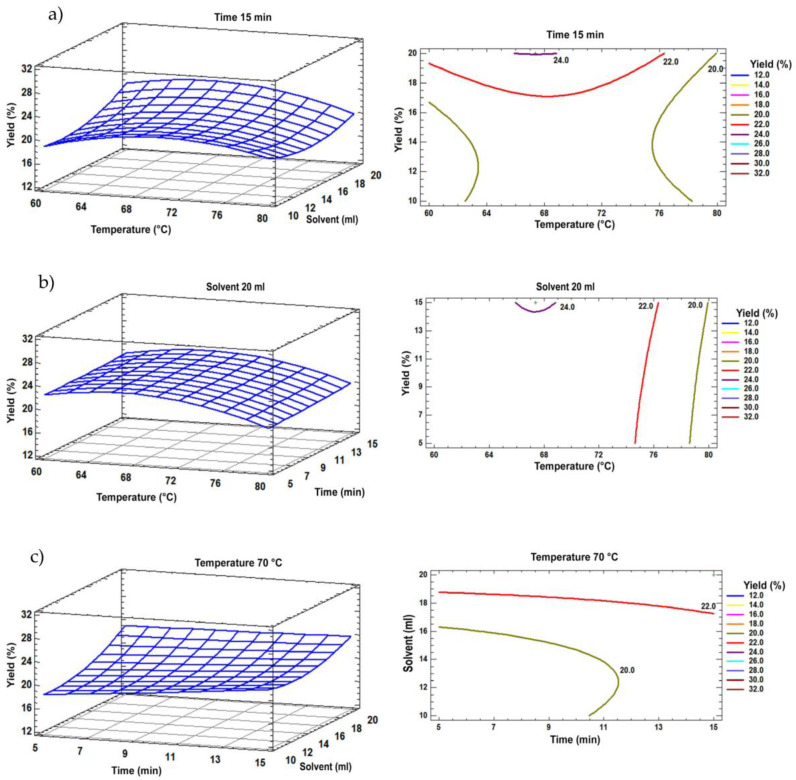
A response surface plot showing the effects of solvent, temperature and processing time obtained by MAE. The presented plots are related to (**a**) time (15 min), (**b**) solvent (20 mL) and (**c**) temperature (70 °C).

**Figure 4 molecules-29-05568-f004:**
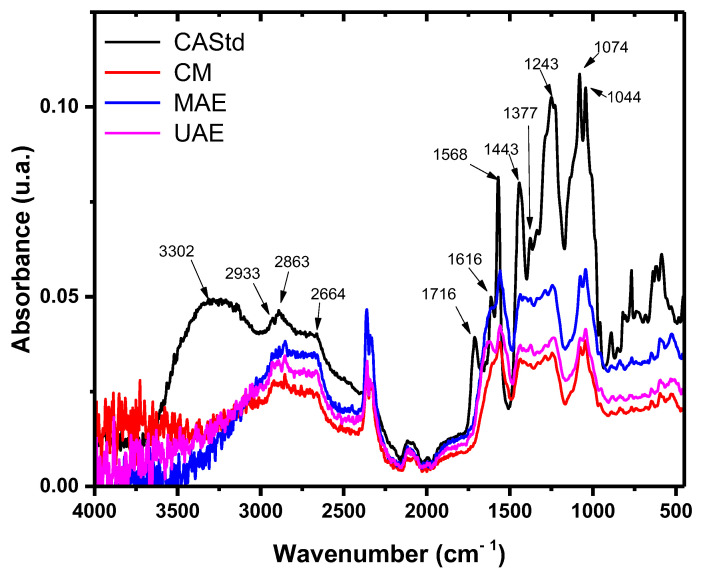
The infrared spectra of the carminic extracts obtained by the different methods. The abbreviations are the following: CAStd: carminic acid standard, MAE: Carminic Extract Microwave, UAE: Carminic Extract Ultrasound and CM: Carminic Extract Conventional Method.

**Figure 5 molecules-29-05568-f005:**
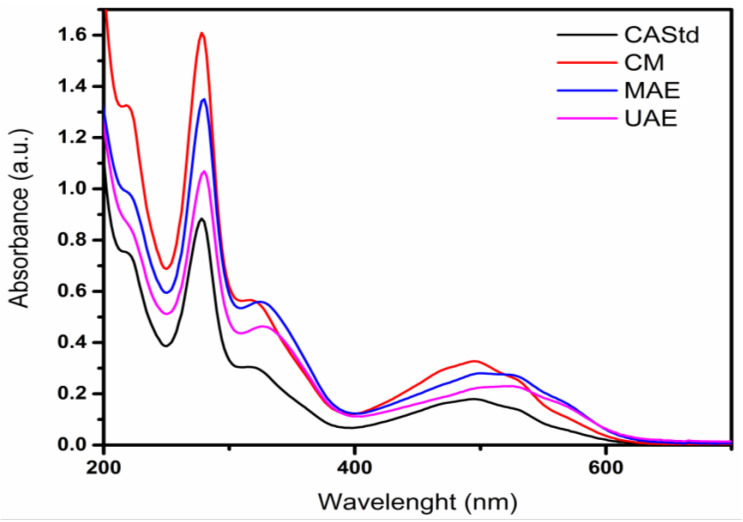
The UV–vis spectra of the obtained cochineal extracts using the conventional (CM), microwave (MAE) and ultrasound (UAE) methods compared to the standard spectrum (CAStd).

**Table 1 molecules-29-05568-t001:** The coefficients of the lineal model yields obtained from the cochineal extractions by ultrasound and microwave techniques using a 2^K^ factorial design.

Run	X1 Temperature (°C)	X2 Irradiation Time (min)	X3 Solvent Volume (mL)	Extract Yield of Cochineal (%) by Ultrasound	Extract Yield of Cochineal (%) by Microwave
1	60	5	15	48.225 ± 5.47	30.922 ± 1.06
2	60	15	15	35.563 ± 0.32	40.478 ± 2.43
3	80	5	15	43.021 ± 5.01	30.078 ± 1.98
4	80	15	15	39.790 ± 0.35	40.889 ± 2.96
5	70	5	10	38.718 ± 3.46	39.578 ± 1.88
6	70	5	20	44.286 ± 2.58	39.400 ± 1.19
7	70	15	10	40.081 ± 1.48	33.789 ± 0.84
8	70	15	20	38.030 ± 8.61	32.789 ± 1.04
9	60	10	10	40.413 ± 3.92	35.147 ± 4.68
10	80	10	10	40.946 ± 6.88	37.300 ± 3.92
11	60	10	20	42.029 ± 6.91	38.333 ± 1.22
12	80	10	20	39.713 ± 1.90	35.611 ± 0.88
13	70	10	15	49.211 ± 3.25	35.667 ± 1.55
14	70	10	15	47.752 ± 2.94	38.622 ± 1.42
15	70	10	15	47.432 ± 2.56	35.756 ± 5.32

The values are the mean ± standard deviation (*n* = 3).

**Table 2 molecules-29-05568-t002:** ANOVA tests of cochineal extractions assisted by ultrasonication and microwaves.

		ANOVA of Ultrasound				ANOVA of Microwave		
Source	S.M.	C.M.	F-Value	*p*-Value	SM	C.M.	F-Value	*p*-Value
A: Temperature	159.24	159.24	9.79	* 0.0036	20.47	20.47	3.08	0.0886
B: Time	19.84	19.84	1.22	0.2773	1.337	1.337	0.20	0.6567
C: Solvent	284.68	284.68	17.51	* 0.0002	376.83	376.83	56.68	* 0.0000
AA	137.88	137.88	8.48	* 0.0064	3.698	3.69	0.56	0.4610
AB	10.87	10.87	0.67	0.4192	3.484	3.48	0.52	0.4742
AC	18.60	18.605	1.14	0.2925	10.64	10.640	1.60	0.2147
BB	0.047	0.047	0.00	0.9572	0.092	0.093	0.01	0.0906
BC	0.977	0.979	0.06	0.8076	20.80	20.803	3.13	0.0862
CC	8.39	8.393	0.52	0.4775	8.86	8.860	1.33	0.2566
Total error	536.52	16.258			219.41	6.648		
Total (corr.)	1309.00				670.55			

S.M. sum of squares, C.M. Mean square. Coefficient values were significantly different when these were followed by * (*p* ≤ 0.05).

**Table 3 molecules-29-05568-t003:** Yields and efficiencies reported for the carminic acid extraction.

Technique	Extract Yield (%)	Extract Efficiency (mg/min)	Carminic Acid Yield (%)	Carminic Acid Efficiency (mg/min)
Conventional Method	31.9	10.6	17.8	2.0
Pressurized Liquids	42.4	14.1	NR	NR
Supercritical Fluids	39.4	1.6	NR	NR
Microwaves	41.0	27.3	18.0	36.0
Ultrasound	49.2	32.8	26.3	17.5

NR: Non Reported.

**Table 4 molecules-29-05568-t004:** FTIR characteristic wavelengths attributed to carminic acid.

Absorption Bands Reported by Cañamares (REF) (cm^−1^)	Absorption Bands Reported in This Work (cm^−1^)	Assignments
3423	3302	ν(OH)
2917	2933	ν(CH_3_)/ν_Glu_(CH)
1716	1716	ν_acid_(C=O)
1617	1616	ν(C=O)/ν_I_(C–C)
1572	1568	ν_I_(C–C)/δ(C_5_OH)/δ(CH)
1376	1377	δ_Glu_(CH)/δ_Glu_(COH)/δ(COH)
1255	1243	ν(C–C)/δ_acid_(COH)/δ(COH)/δ_Glu_(CH)
1083	1074	ν_Glu_(C–O)/δ_Glu_(COH)
1045	1044	ν_Glu_(C–C)/δ_Glu_(COH)
820	813	γ(COH)

ν = stretching; δ = in-plane bending; γ = out of plane bending; Glu = glucose residue. νI = stretching vibrations occurring in benzene ring.

**Table 5 molecules-29-05568-t005:** Range of examined parameters for 2^K^ Factorial design.

Independent Variables		Levels Codification		
		−1	0	1
Temperature (°C)	X1	60	70	80
Irradiation time (min)	X2	5	10	15
Solvent ratio (mL)	X3	10	15	20

Treatments were generated with Statgraphics Centurion XVI software version 16.2.04.

## Data Availability

Data are contained within the article.

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
