# Peer review of "Cochineal (Dactylopius coccus Costa) Pigment Extraction Assisted by Ultrasound and Microwave Techniques"

_molecules, 2024, doi:10.3390/molecules29235568_

Round 1

Reviewer 1 Report

Comments and Suggestions for Authors

The paper presents the extraction of cochineal pigments employing ultrasound and microwave assisted methodologies. The subject is of scientific interest but the authors have focused the work on optimizing the extraction conditions using a factorial design approach, nevertheless I think the work should be completed by studying the selectivity of the extraction and the characterization of the extracts using other analytical techniques. Thus I think it could be published after a major revision.

·         Introduction: A deeper revision and discussion of the published works related to the carminic acid extraction should be performed.

·         Line 83: “temperature, irradiation time and solvent volume, and the parameters condition (levels) was based on the knowledge acquired previously” this should be supported with a reference.

·         Line 202: “also flavokermesic and kermesic acid, which have been reported by Gonzales et al 2002”. The reference number should be included.

·         Line 210: “The carminic acid extraction assisted by ultrasound and microwave as green technologies provide advantages over the conventional extraction method, since they do not modify the molecular composition”. In order to have a better knowledge about the chemical composition of the extract, HPLC-MS analysis should be performed.

·         Selectivity of the extraction should be assessed using HPLC analysis. Not all the compounds extracted are carminic acid.

·         How was the extraction yield calculated?

·         Ref 20 is not related to the work

Author Response

Thank you very much for taking the time to review this manuscript. Please find the detailed responses below and the corresponding corrections 

Comments 1.- A deeper revision and discussion of the published works related to the carminic acid extraction should be performed.

Response 1.-Thank you for your observation, the change was made.

Comments 2.- Line 83: “temperature, irradiation time and solvent volume, and the parameters condition (levels) was based on the knowledge acquired previously” this should be supported with a reference.

Response 2.- Thank you for your observation, the reference was added (Page 13, line 354).

Comments 3.- Line 202: “also flavokermesic and kermesic acid, which have been reported by Gonzales et al 2002”. The reference number should be included.

Response 3.- Thank you for your observation, the change was made (Page 10, line 218).

Comments 4.- Line 210: “The carminic acid extraction assisted by ultrasound and microwave as green technologies provide advantages over the conventional extraction method, since they do not modify the molecular composition”. In order to have a better knowledge about the chemical composition of the extract, HPLC-MS analysis should be performed.

Response 4.- In this section we refer to the molecular composition of carminic acid, which was characterized by FTIR where we can observe the bands attributed to anthraquinones, sugar and carboxyl group. We appreciate the suggestion to use the HPLC-MS technique, however, the determination by the techniques used in this work is more than sufficient to determine the identity of the compound obtained by comparison against commercial standards, prioritizing the use of lower cost and faster techniques for their subsequent implementation in industrial conditions. It is important to mention that normally the carmine extract usually contains a mixture of carminic acid and other compounds synthesized during its metabolic pathway, such as kermesic acid, flavokermesic acid, among others. This can also be verified by using commercial standards with a purity grade of 96 %, showing a spectrum similar to the one obtained in this work and from previous studies. Further work to verify not only the identity but also the variants of the compounds obtained will be contemplated.

Comments 5.- Selectivity of the extraction should be assessed using HPLC analysis. Not all the compounds extracted are carminic acid.

Response 5.- The UV-Vis analytical technique shows that we do not have pure carminic acid as shown in Figure 5, where different absorption bands attributed to the other compounds ranging from 400 to 600 nm can be found. However, the objective of this work was the use of these green, eco-friendly technologies for the extraction of this pigment and not its purification since it is a mixture that is used in the industry in this way, so purifying it would require greater processes that would make it more expensive and generate pollutants (González, 2002; Ferreyra-Suárez, 2024).

Comments 6.- How was the extraction yield calculated?

Response 6.- The explanation of the paragraph has been modified (Page 13, lines 341 to 347).

Comments 7.- Ref 20 is not related to the work

Response 7.- Thank you for your observation, the change was made (Page 11, line 282).

Reviewer 2 Report

Comments and Suggestions for Authors

The manuscript focuses on the isolation and characterization of carminic acid from Dactylopius coccus Costa. The pigment extracts were obtained by ultrasound- and microwave-assisted extraction which process optimization and efficiencies were compared. The title of the paper adequately reflects the subject under investigation in the proposed study. Text layout is preserved in accordance with the requirements of the editorial. The manuscript is very well written with a good quality of presentation. The language and structure of the article are clear and easily understandable. I found this manuscript to be of interest for both scientific community and industry. However, some issues need to be addressed:

1.        More key-findings need to be added in the Abstract section. Overall conclusion is missing.

2.        The importance of carminic acid and should be more highlighted in the Introduction section.

3.        Line 83-101. The text should be moved to the subsection 4.9.

4.        SD values should be reported for selected responses.

5.        The following abbreviations should be changed Extraction Assisted by Ultrasound (EAU) to Ultrasound-assisted extraction (UAE) and Extraction Assisted by Microwave (EAM) to Microwave-assisted extraction (MAE).

6.        Conclusion section should be more supported by the results. Practical application and future perspective should be included in the Conclusions section.

7.        The authors should check the text for typos and grammatical errors.

Author Response

Thank you very much for taking the time to review this manuscript. Please find the detailed responses below and the corresponding corrections 

Comments 1.- More key-findings need to be added in the Abstract section. Overall conclusion is missing.

Response 1.- Thank you for your observation, the change was made (Page 1, lines 22 to 40).

Comments 2.- The importance of carminic acid should be more highlighted in the Introduction section.

Response 2.- Thank you for your observation, the change was made (Page 2, lines 46 to 102 ).

Comments 3.- Line 83-101. The text should be moved to the subsection 4.9

Response 3.- Thank you for your observation, the change was made (Page 4, lines 349 to 359).

Comments 4.- SD values should be reported for selected responses

Response 4.- Thank you for your observation, the change was made 

Comments 5.- The following abbreviations should be changed Extraction Assisted by Ultrasound (EAU) to Ultrasound-assisted extraction (UAE) and Extraction Assisted by Microwave (EAM) to Microwave-assisted extraction (MAE).

Response 5.- Thank you for your observation, the change was made.

Comments 6.- Conclusion section should be more supported by the results. Practical application and future perspective should be included in the Conclusions section.

Response 6.- Thank you for your observation, the change was made (Page 13, lines 366 to 378).

Comments 7.- The authors should check the text for typos and grammatical errors

Response 7.- Thank you for your observation.

Round 2

Reviewer 1 Report

Comments and Suggestions for Authors

The authors have taken into account my advice for the revision of the paper, but there are still some aspects that need to be improved.

Abstract: English grammar should be revised

Line 59: ” cosmetics and textiles due to its high chemical and biological stability, and it [4,5].” This phrase should be revised

Line 65: “which corresponds to the lacquer obtained by the union by the union of carminic acid with salts such as aluminum or calcium [10].” Should be revised

Line 70: “Conventional extraction methods include simple, inexpensive and easy-to-use techniques”. These techniques should be named (solid liquid extractions? which solvents ...)

Table1: which are the new values added, %RSD, SD???. The number of decimal places of the relative standard deviation and the values of yield should be revised. What was the value of n?

Line 160: “reaching a yield of 41%” any decimal places?  RSD value?

Line 347: “CA yields are reported as the % of carminic acid content present in the extract”. This percentage is expressed as mass/volume or mass/mass.  Was the extract evaporated to dryness for the calculation of the extraction yield??

Author Response

Thank you very much for taking the time to review this manuscript. Please find the detailed responses below and the corresponding revisions/corrections 

Comments 1.-  English grammar should be revised

Response 1.- Thank you for your observation, the change was made.

Comments 2.- Line 59:  cosmetics and textiles due to its high chemical and biological stability, and it [4,5].” This phrase should be revised

Response 2.- Thank you for your observation, the change was made.

Line 65: “which corresponds to the lacquer obtained by the union by the union of carminic acid with salts such as aluminum or calcium [10].” Should be revised

Response 3.- Thank you for your observation, the correction has been made.

Line 70: “Conventional extraction methods include simple, inexpensive and easy-to-use techniques”. These techniques should be named (solid liquid extractions? which solvents ...)

Response 4.- Thank you for the observation the change was made.

Table1: which are the new values added, %RSD, SD???. The number of decimal places of the relative standard deviation and the values of yield should be revised. What was the value of n?

Response 5.- Thanks for the comment, the correction has been made.

Line 160: “reaching a yield of 41%” any decimal places?  RSD value?

Response 6.- Thanks for the comment, The value of the optimization of the extract yield by MAE (41.00 %) was obtained by the mathematical analysis of the experimental runs of the box behnken design. It was the optimal result provided by the software.

Line 347: “CA yields are reported as the % of carminic acid content present in the extract”. This percentage is expressed as mass/volume or mass/mass.  Was the extract evaporated to dryness for the calculation of the extraction yield??

Response 7.-  Thank you for the observation. All extracts obtained by the conventional method and ultrasound and microwave assisted extraction were brought to complete dryness using a rotary evaporator at 80 °C under reduced pressure and lyophilized. CA yields are reported as the % of carminic acid (w/w) content present in the extract 
